# A Highly Water-Soluble and Solid State Emissive 1,8-Naphthalimide as a Fluorescent PET Probe for Determination of pHs, Acid/Base Vapors, and Water Content in Organic Solvents

**DOI:** 10.3390/molecules27134229

**Published:** 2022-06-30

**Authors:** Nikolai I. Georgiev, Paoleta V. Krasteva, Ventsislav V. Bakov, Vladimir B. Bojinov

**Affiliations:** Department of Organic Synthesis, University of Chemical Technology and Metallurgy, 8 Kliment Ohridsky Str., 1756 Sofia, Bulgaria; paoletyn@gmail.com (P.V.K.); vencobakov@gmail.com (V.V.B.)

**Keywords:** 1,8-naphthalimide fluorescent probe, solid state emission, photoinduced electron transfer (PET), INH logic gate, acid-base vapors, water content

## Abstract

A new highly water-soluble 1,8-naphthalimide fluorophore designed on the “*fluorophore-spacer-receptor_1_-receptor_2_*” model has been synthesized. Due to the unusually high solubility in water, the novel compound proved to be a selective PET-based probe for the determination of pHs in aqueous solutions and rapid detection of water content in organic solvents. Based on the pH dependence of the probe and its high water solubility, the INH logic gate was achieved using NaOH and water as chemical inputs, where NaOH is the disabler and the water is an enabler. In addition, the probe showed effective fluorescence “*off-on*” reversibility on glass support after exposure to acid and base vapors, which defines it as a promising platform for rapid detection of acid/base vapors in the solid-state, thus extending the molecular sensing concept from solution to the solid support.

## 1. Introduction

In the past years, the development of fluorescent sensors and fluorescent sensing materials has been a topic of research in chemistry [1,2,3,4,5]. The fluorescence signaling output attracts due to several advantages such as cheap equipment with high sensitivity, immediate response, harmless and non-invasiveness suitable for real-time bioimaging and diagnostic medicine [6,7,8,9,10,11]. Consequently, major attention has been focused on the design and synthesis of fluorescent probes for a variety of analytes [12,13,14,15]. The used approaches were based on intramolecular charge transfer (ICT), photoinduced electron transfer (PET), twisted intramolecular charge transfer (TICT), fluorescence resonance energy transfer (FRET), excited-state intramolecular proton transfer (ESIPT), and aggregation-induced emission [16,17,18,19,20,21,22,23,24]. Notably, the PET using the “*fluorophore-spacer-receptor*” format was recognized as the most popular platform for the design of fluorescence chemosensing probes [25,26,27,28]. This model was distinguished by simple construction and easier and predictable communication between the receptor (recognition part) and the fluorophore (signaling part). As a result, the PET process was recently tested extensively in the most common fluorophores and a large number of PET-based probes were reported. In addition, the PET process and the “*fluorophore-spacer-receptor*” format were involved even in the more complicated molecular logic devices for multicomponent analysis [29,30,31,32]. However, the synthesis of PET chemosensing probes with improved properties and better applicability is still a major task.

Generally, the organic fluorescent probes are well soluble in organic solvents and practically are insoluble in water. At the same time, the chemical analysis in the solution is mostly performed in water. Additionally, water-soluble probes were preferred for real-time monitoring of living objects due to their lower toxicity and higher biocompatibility. That is why in the last years, more and more attention has been paid to the design and synthesis of fluorescent probes with higher water solubility [33,34,35,36,37].

Herein we are reporting on a highly water-soluble probe based on 1,8-naphthalimide fluorophore (Figure 1). This work is an extension of our previous report, where two amidoamine containing 1,8-naphthalimides with water solubility in a concentration of 10^−5^ mol/L was described [38]. In contrast, the novel compound showed higher water solubility resulting in stable water solutions even at concentrations 10^−2^ mol/L. Furthermore, due to the unusual solubility in water, the synthesized 1,8-naphthalimide was successfully applied as a PET sensing probe for the detection of water content in organic solvents.

The rapid detection of water in organic solvents is very important in large areas such as industrial production, food processing, and biomedical and environmental monitoring [39,40,41,42]. Nevertheless, the PET probes for water content are unusual and rare [43,44,45]. Hence, the synthesis of PET fluorescent probes for water content currently could be of great interest. In addition, the novel compound showed PET fluorescence sensing properties in a solid-state toward acid/base vapors. Extending the molecular sensing concept from solution to solid support currently is a major task, which opens possibilities in new directions and practical applications [46,47,48]. However, only a few PET sensing compounds were reported as solid-state emissive probes [49,50]. That is why we believe that the present work should be of theoretical and practical significance in PET chemosensing studies for the development of novel, simple, and low-cost PET-based sensors for rapid detection of pH, acid/base vapors, and water content in organic solvents.

## 2. Results and Discussion

### 2.1. Design and Synthesis

The novel probe **4** was designed on a classic “*fluorophore-spacer-receptor*” model, as the methylpiperazine unit containing two electron-rich amino groups is the receptor and the 1,8-naphthalimide unit is the fluorophore. The introduction of amines in the *N*-position of the 1,8-naphthaimides significantly increases their hydrophilicity [51,52,53]. That is why to obtain a water-soluble compound, the piperazine receptor in compound **4** was bound to the *N*-position of the fluorophore. We chose an unsubstituted 1,8-naphthaimide unit instead of the commonly used 4-amino or 4-oxy substituted one due to the push-pull ICT nature of the 1,8-naphthaimide derivatives. It is well known that in the 1,8-naphthaimide exited state occurs charge transfer from the C-4 electron-donating position to the carbonyl electron-accepting groups that generate an electron-rich field around the imide [54,55]. The strong repulsive character of the resulted field seriously restricts the PET process from *N*-position to the fluorophore in the electron-rich architecture such as 4-amino-1,8-naphthalimides [56,57,58]. This effect was reduced in the unsubstituted electron poorer derivatives that generated a weaker repulsive field around the imide cycle of the fluorophore [59].

The synthesis of the PET-based 1,8-naphthalimide probe **4** is illustrated in Figure 2. It was performed in three steps using 1,8-naphthalic anhydride as a starting material. In the first step, equimolar amounts of hydrazine monohydrate and 1,8-naphthalic anhydride were condensed in a methanol solution. Then the so-prepared *N*-amino-1,8-naphthalimide **2** was reacted with chloroacetyl chloride at 70 °C in dioxane for 3 h to afford intermediate **3**. In the last step, the desired compound **4** was obtained after a nucleophilic substitution of the chlorine in the intermediate **3** with a methylpiperazine moiety. The reaction was performed in boiling DMF for 2.5 h.

### 2.2. Chemosensing Properties in Aqueous Solution

#### 2.2.1. pH Sensing Properties

The synthesized probe **4** showed unusually high water solubility for the 1,8-naphthalimide derivatives. It was found that clear and stable probe solutions are formed even at a concentration of 5 × 10^−2^ mol/L. This encouraged us to investigate the photophysical behavior of the probe in a pure water solution. Due to the ICT nature of the 1,8-naphthalimide unit, compound **4** showed absorption long-wavelength band from 290 nm to 380 nm with maximal intensity at 344 nm (Figure 1) and fluorescence signaling output in the range between 350 nm and 480 nm with a maximum at 397 nm (Figure 2) [38,53]. The observed absorption spectra were pH-independent, which is illustrated in Figure 1, where the absorption spectra of probe **4** at pH 3, pH 7, and pH 11 are depicted as an example. In contrast to the absorption spectra, high pH-dependent fluorescence intensity was observed in the corresponding fluorescent spectra. This behavior was expected and attributed to the PET phenomenon, which occurs only in the excited state and does not affect the fluorophore ground state, respectively, its absorption spectra.

Because of the PET process from the electron-rich piperazine amines to the excited state of the 1,8-naphthalimide unit, the fluorescence emission of compound **4** was completely quenched in an alkaline media. Upon protonation of the piperazine amines in an acid media, the fluorescence quenching PET was impossible, and compound **4** showed bright fluorescence. Owing to the stepwise protonation of both piperazine amines, the observed pH titration plot of probe **4** consisted of two S-shaped curves (Inset of Figure 2).

According to the titration plot in Figure 2, the first piperazine amine protonation occurred in a pH window of 12–7.7, that logically could be attributed to the quaternization of the methylamino group possessing electron-donating methyl fragment (Figure 3).

From the obtained curve, a p*K*_a_ value of 9.09 for the first protonation of probe **4** was calculated using Hendersen–Hasselbalch Equation (1) [60]. This value is consistent with the reported tertiary alkylamine p*K*_a_ values.
(1)$$\begin{array}{c} \mathrm{pH}=pK_{a}+\log\frac{\left(I_{max}-I\right)}{\left(I-I_{min}\right)} \end{array}$$
where *I*_*max*_ and *I*_*min*_ are the maximum and minimum fluorescence intensity, respectively, and *I* is the fluorescence intensity at the corresponding pH value.

Furthermore, Figure 2 reveals a second quaternization process in the piperazine moiety of probe **4** occurring in a pH window of 7.7–3, which analysis according to Equation (1) gives p*K*_a_ = 7.33. This p*K*_a_ value is slightly lower than the usual values for tertiary alkylamines and could be attributed to the protonation of the methylene-bound amine in **4**. The methylene amine in **4** is connected with the electron-withdrawing amide that decreased the electron density around neighboring groups and increased their acidity.

The influence of the most common cations and anions on the fluorescence output of the novel probe **4**, such as Co^2+^, Cu^2+^, Fe^3+^, Ni^2+^, Pb^2+^, Cd^2+^, Zn^2+^, Hg^2+^, Cl^−^, NO_3_^−^, SO_4_^2−^, HSO_4_^−^, CO_3_^2−^, CH_3_COO^−^, Br^−^, NO_2_^−^, SO_3_^2−^, PO_4_^3−^, and F^−^ were also tested as potential analytes. The study was performed at pH 4.5, pH 7.3, pH 8, and pH 10. However, in all cases, the tested ions (10^−4^ mol/L) caused only a minor quenching of the probe’s fluorescence intensity. This observation shows that the examined compound **4** could be used as a selective probe for the determination of pHs in aqueous solutions.

#### 2.2.2. Detection of Water Content in Organic Solvents

The p*K*_a_ values of the tertiary alkyl amines are usually about p*K*_a_ ≈ 9 [61,62,63]. At the same time, under normal conditions, the water pHs are in a range of pH 6~pH 8, which allows protonation of the classical PET alkylamino receptors. This statement makes the PET sensors a promising platform for the design of fluorescent probes for the detection of water content in organic solvents. However, most PET probes are water-insoluble, which results in a mixed signaling output—fluorescent enhancement due to the prevention of the PET process and fluorescent quenching due to the aggregation caused by quenching. That is why the reports of PET probes for the detection of water content in organic solvents are very rare and unusual. The recent research progress in PET sensors is limited to a few number of probes for the detection of water content in organic solvents using amino acids as a receptor fragment [64,65,66].

Similarly, the synthesized compound **4** is highly water-soluble and possesses a PET receptor containing both an amino group and an acid fragment (acid amide). This encouraged us to investigate the ability of probe **4** to detect water content in organic solvents. For this purpose, the fluorescent spectra of **4** were recorded in dry ethanol and mixtures of ethanol/water (Figure 3A). As can be seen from Figure 3, the fluorescent intensity of probe **4** increased gradually with the increase in water content in the solutions under study. This is due to both the changes in pH and the microenvironment around the fluorophore. Since the main effect is pH, which is well defined only in pure water, the solution pH value is strongly dependent on the water content in the organic solvent. From the fluorescent changes at 390 nm, a calibration plot was constructed, which showed a linear range from 4% to 40% of water content (Figure 3B). Based on the linear calibration plot, the limit of detection LOD = 3% was calculated using the equation LOD = 3 σ/b, where b is the slope and σ is the standard deviation of 10 measurements in the dry solvent [67].

#### 2.2.3. Molecular Logic

To confirm our statement that the observed fluorescence enhancement of probe **4** in the presence of water was due to a blocked PET process after protonation of the tertiary alkyl amines in the probes PET receptor fragment, the fluorescence spectra of probe **4** were investigated in mixtures of ethanol and water containing 10^−6^ mol/L NaOH or NH_3_. As we expected, a fluorescent enhancement was not observed due to the lowered acidity of the water, which is not able to prevent the PET process in probe **4** under these conditions. Based on this behavior of compound **4**, an INH molecular logic gate was constructed (Figure 4).

As chemical inputs in this logic gate, NaOH (10^−6^ mol/L) and water (50% content) in an ethanol solution containing 10^−5^ mol/L of probe **4** were used. The obtained signaling output of the achieved logic gate was depicted in Figure 4. As can be seen, higher emission of **4**, coded in binary as 1, is observed only in the presence of water as an input, while the presence of NaOH blocks the effect of water and the system shows low fluorescence, coded in binary as 0, in all other input combinations. This behavior mimics very well the INH logic gate, where the water is the enabler and NaOH is the disabler [68,69,70].

#### 2.2.4. Detection of Acid/Base Vapors in Solid State

In our recent research, we have found that thin films of some classical PET-based 1,8-naphthalimides could be used effectively for the detection of acid or basic vapors due to the switching between PET process and solid-state emission in the signaling fluorophore after exposure of acid/base vapors [49]. This was the reason for preparing a thin film of compound **4** on a glass support after spraying of ethanol solution of **4** onto a glass plate and evaporation of the solvent in air. The resulted film was exposed for 2 s to HCl and then to NH_3_ vapors. The glass samples were photographed (Figure 5) and their fluorescence spectra were recorded after each exposure (Figure 6A).

Before exposure, the fluorescence emission of the examined film showed low fluorescence due to the possible PET quenching process in probe **4**. After the exposure to HCl vapors, the PET process was disallowed and the fluorescence was amplified, which was visible even by a naked eye (Figure 5). Furthermore, the high fluorescence output of **4** was “*turned off*” again after exposure to NH_3_ vapor for 2 s (Figure 6A) due to the deprotonation of the fluorescent quaternary ammonium salt of compound **4**.

The observed fluorescence enhancement (FE) was calculated to be FE = 10. Also, it was found that the studied thin film of probe **4** could be transferred between “*off*” and “*on*” states reversibly at least seven times without changes in the fluorescence intensity in both “*off*” and “*on*” states (Figure 6B). These results clearly showed that compound **4** could be used as an efficient platform for rapid detection of acid/base vapors in the solid-state.

## 3. Materials and Methods

### 3.1. Materials

The starting commercial reagents 1,8-naphthalic anhydride (Sigma-Aldrich Product No.: N1607; CAS No.: 81-84-5; EC No.: 201-380-2, Sigma-Aldrich Co., St. Louis, MO, USA), hydrazine monohydrate 99 + % (Fisher Scientific Product No.: AA1665136; CAS No.: 7803-57-8, Fisher Scientific, Waltham, MA, USA), 1-methylpiperazine 99% (Sigma-Aldrich Product No.: 130001; CAS No.: 109-01-3; EC No.: 203-639-5, Sigma-Aldrich Co., St. Louis, MO, USA) and chloroacetyl chloride 98% (Sigma-Aldrich Product No.: 104493; CAS No.: 79-04-9; EC No.: 201-171-6, Sigma-Aldrich Co., St. Louis, MO, USA) were used without purification. The intermediate compounds *N*-amino-1,8-naphthalimide **2** and *N*-Chloroacetamide-1,8-naphthalimide **3** were synthesized as we described before [38]. The Sigma-Aldrich salts at p.a. grade Hg(NO_3_)_2_.H_2_O (CAS No.: 7783-34-8), Cu(NO_3_)_2_.H_2_O (CAS No.: 10031-43-3), Zn(NO_3_)_2_.6H_2_O (CAS No.: 10196-18-6), Ni(NO_3_)_2_.6H_2_O (CAS No.: 13478-00-7), Co(NO_3_)_2_.6H_2_O (CAS No.: 10026-22-9), Cd(NO_3_)_2_.4H_2_O (CAS No.: 13477-34-4), Fe(NO_3_)_3_. 9H_2_O (CAS No.: 7782-61-8) and Pb(NO_3_)_2_ (CAS No.: 10099-74-8) were used as a source of metal cations. The Sigma-Aldrich salts at p.a. grade KCl (CAS No.: 7447-40-7), NaNO_3_ (CAS No.: 7631-99-4), Na_2_SO_4_ (CAS No.: 7757-82-6), NaHSO_4_ (CAS No.: 7681-38-1), Na_2_CO_3_ (CAS No.: 497-19-8), CH_3_COONa (CAS No.: 127-09-3), KBr (CAS No.: 7758-02-3), NaNO_2_ (CAS No.: 7632-00-0), Na_2_SO_3_ (CAS No.: 7757-83-7), K_3_PO_4_.H_2_O (CAS No.: 27176-10-9) and NaF (CAS No.: 7681-49-4) were used as a source of anions.

### 3.2. Methods

FT-IR spectra were recorded on a Thermo Scientific Nicolet iS20 FTIR spectrometer (Thermo Fisher Scientific, Waltham, MA, USA). The ^1^H NMR analysis was performed on a Bruker AV-600 spectrometer with an operating frequency of 600 MHz. Electrospray ionization mass spectra (ESI-MS) were obtained on a BRUKER micrOTOF-Q system. The TLC monitoring was performed on silica gel, ALUGRAM^®^SIL G/UV254, 40 × 80 mm, 0.2 mm silica gel 60. A Hewlett Packard 8452A spectrophotometer was used for the UV-VIS absorption measurements. The photophysical study was performed at room temperature (25.0 °C) in 1 × 1 cm quartz cuvettes. The fluorescence spectra were recorded using a Scinco FS-2 spectrofluorimeter. The quantum yields of fluorescence (ΦF) were calculated relatively to 9,10-Diphenylanthracene (ΦF = 0.95 in ethanol) [71]. The HANNA^®^instruments HI-2211 Bench Top pH meter was used in the pH monitoring. A very small amount of hydrochloric acid and sodium hydroxide was used to adjust the pH. The influence of metal cations and anions on the fluorescence emission was studied by adding portions of ion stock solution to 10 mL of the fluorophore solution. The addition was limited to 100 μL so that dilution remains insignificant. The ions were added gradually up to 10 equivalents (10^−4^ mol/L) to a fluorophore solution (10^−5^ mol/L). The effect of ions was studied at constant pH in the presence of 10 µM phosphate (pH 6), 10 µM HEPES (pH 7.2), 10 µM Tris-HCl (pH 8), or 10 µM ammonia/ammonium chloride (pH 10) buffer solutions.

### 3.3. Synthetic Procedures

#### 3.3.1. Synthesis of *N*-Amino-1,8-Naphthalimide **2**

A hydrazine monohydrate (0.5 mL, 0.01 mol) was added to a solution of 1,8-naphthalic anhydride (2 g, 0.01 mol) in 30 mL of methanol. The resulting mixture was heated under reflux for 2 h. After cooling, the precipitate was filtered off, washed with methanol, and dried to give pale yellow crystals of *N*-amino-1,8-naphthalimide (1.5 g, 71%). FT-IR (KBr) cm^−1^: 3310 and 3233 (*ν* NH_2_); 1704 (*ν*^as^ N-C=O); 1648 (*ν*^s^ N-C=O).

#### 3.3.2. Synthesis of *N*-Chloroacetamide-1,8-Naphthalimide **3**

1.5 g of *N*-amino-1,8-naphthalimide **2** (0.007 mol) were added to 7 mL of dry dioxane. The resulted suspension was heated to 70 °C and then 2.8 mL of chloroacetyl chloride (0.035 mol) was added dropwise. The mixture was stirred at the same temperature for 3 h and the solid that precipitated after cooling was filtered off to give intermediate **3** as white crystals (0.88 g, 45%). ^1^H NMR (CHCl_3_-*d*, 600.13 MHz) *δ* ppm: 9.00 (s, 1H, *NH*CO); 8.58 (d, 2H, *J* = 7.3 Hz, naphthalimide H-2 and H-7); 8.22 (d, 2H, *J* = 8.2 Hz, naphthalimide H-4 and H-5); 7.73 (t, 2H, *J* = 7.7 Hz, naphthalimide H-3 and H-6); 4.28 (s, 2H, CO*CH*_2_Cl).

#### 3.3.3. Synthesis of Probe **4**

To the solution of *N*-chloroacetamide-1,8-naphthalimide **3** (0.88 g, 0.003 mol) in 5 mL of DMF, 1.4 mL of methylpiperazine (0.012 mol) were added. Then the resulted solution was heated under reflux for 2.5 h. The precipitated solid after cooling was filtered off and dried to give white crystals of the desired probe **4** (0.54 g, 60%). FT-IR (KBr) cm^−1^: 3357 (*ν* NH); 1725 (*ν* N-C=O); 1698 (*ν*^as^ N-C=O); 1662 (*ν*^s^ N-C=O). ^1^H NMR (D_2_O, 600.13 MHz) ppm: 8.29 (d, 2H, *J = 7.3* Hz, naphthalimide H-2 and H-7), 8.13 (d, 2H, *J = 8.1* Hz, naphthalimide H-4 and H-5), 7.61 (t, 2H, *J = 7.7* Hz, naphthalimide H-3 and H-6), 3.67 (s, 2H, -CO*CH*_2_-), 3.63 (m, 2H, *CH*_2_ piperazine), 3.36 (m, 2H, *CH*_2_ piperazine), 3.31 (m, 2H, *CH*_2_ piperazine), 2.98 (s, 3H, *CH*_3_), 2.84 (m, 2H, *CH*_2_ piperazine). Elemental analysis: Calculated for C_19_H_20_N_4_O_3_ (MW 352.39) C 64.76, H 5.72, N 15.90%; Found C 64.83, H 5.68, N 16.05%. Positive-ion ESI-MS at *m*/*z*: 353.0176 [M + H]^+^.

## 4. Conclusions

In summary, we presented here the synthesis of the novel 1,8-naphthalimide fluorophore as a selective PET-based probe for the determination of pH in aqueous solutions, designed on a “fluorophore-spacer-receptor1-receptor2” architecture. Because of the PET process from the electron-rich piperazine amines to the excited state of the 1,8-naphthalimide unit, the fluorescence emission of the probe was completely quenched in alkaline media, while upon protonation of the piperazine receptors, the probe restored its bright fluorescence. Owing to the stepwise quaternization of both piperazine amines, two S-shaped curves with p*K*_a_ values of 9.09 and 7.33, corresponding to lower and higher fluorescent intensities, respectively, were calculated. Due to the unusual high solubility in water, the synthesized compound was successfully applied as a PET sensing probe for the detection of water content in organic solvents with a limit of detection LOD = 3%. In addition, using NaOH and water as chemical inputs, an INH logic gate was achieved, where the water is the enabler and NaOH is the disabler. The probe also showed reversible “*off*-*on*” PET-dependent fluorescence on solid support when exposed to acid and base vapors. The studied thin film of the probe was found to be able to switch between “*off*” and “*on*” states reversibly at least 6 times without changes in the fluorescence intensity in both “*off*” and “*on*” states. The results obtained clearly show that the novel compound could be used as an efficient solid-state emissive probe.

## Data Availability

The authors declare that the data supporting the findings of this study are available within the article.

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
