# Peer review of "A Highly Water-Soluble and Solid State Emissive 1,8-Naphthalimide as a Fluorescent PET Probe for Determination of pHs, Acid/Base Vapors, and Water Content in Organic Solvents"

_molecules, 2022, doi:10.3390/molecules27134229_

Round 1

Reviewer 1 Report

The said manuscript by Bojinov et al describes synthesis of water soluble sensor for detection of acid and base. Although fluorescence changes were observed with addition of acid and base but 

Major changes are required.

1. The language used in the manuscript needs major improvement.  

2. With increase in water content, fluorescence enhancement has been observed but no reason for this change has been given. Explain the reason for same.

3. Explain the UV vis changes with addition of acid and base.

On solid support, acid and base vapor addition shows reversibility. Explain for how many cycles, this reversibility could be observed.

Author Response

1) As Reviewer #1 recommends, the language used in the revised version of the manuscript is improved.

2) According to the recommendation of Reviewer #1, short explanation for the fluorescence enhancement with the increase in water content is added to section 2.2.2. Detection of Water Content in Organic Solvents (Paragraph 2, line 7) of the revised version of the manuscript.

3) According to the remark of Reviewer #1, elucidative text for the UV/Vis changes with addition of acid and base is added to section 2.2.1. pH Sensing Properties (Paragraph 1, line 8) of the revised version of the manuscript.

4) According to the recommendation of Reviewer #1, explanation for how many cycles, this reversibility could be observed is illustrated by addition of a new Figure 6B (vbbojinov_Figure 6BR1) to the revised version of the manuscript. The former Figure 6 is renamed to Figure 6A (vbbojinov_Figure 6AR1). The caption of Figure 6 is also replaced with a new one.

Figure 6. (A) Fluorescence spectra of a solid film of probe 4, exposed first on HCl and then on NH3 vapors (λex = 340 nm); (B) Cycle Index of a solid film of probe 4.

Reviewer 2 Report

The paper is clear and well written.

The topic of the paper is appropriate for the Journal.

The text of the paper was written correctly in terms of stylistically, punctuation and terminology.

The paper was correctly edited and graphically developed at a good level.

The literature was chosen correctly and fully used in the paper.

I do not see any shortcuts.

The paper deserves a positive assessment because it is current and interesting from both a cognitive and practical point of view.

Author Response

We thank the Reviewer #2 for the positive comments on our work.

Reviewer 3 Report

Please address the following comments/suggestions.

The work is interesting and useful. However, the following are the minor comments.

(i)
Grammar and punctuations should be rechecked and rectified, a few problems are catching the eye as there are several grammatical mistakes in the manuscript which should be improved, English correction is very essential throughout the manuscript.

 (ii) Author should mention the actual purity of the chemicals given by the company. The precursors and all the chemicals used during the research work should be given including their CAS number and other necessary details.

(iii) Maximum wavelength of UV/Vis absorption spectra should be compared with the literature, If possible

(iv) In figure 3B, there is a comma in regression value which should be written as point “0.9977”

(v) Please rewrite or refine the conclusion by giving exact findings from the results and discussion.

Author Response

1) As Reviewer #3 recommends, the language used in the revised version of the manuscript is improved. The grammar and punctuations are rechecked and rectified.

2) As Reviewer #3 recommends, the actual purity of the precursors and all the chemicals used during the research work are given in section 3.1. Materials (Paragraph 1) of the revised version of the manuscript including their CAS numbers.

3) It is very difficult to compare the photophysical characteristics of the probe, including UV/Vis absorption data, because it is well known that the photophysical properties of the 1,8-naphthalimide derivatives depend mainly on the polarization of their chromophoric system. The latter is a function of the C-4 substituents and the polarity of the medium. Moreover, the high water solubility for the 1,8-naphthalimide derivatives is very unusual. Nevertheless, in section 2.2.1. pH Sensing Properties (Paragraph 1, line 7) of the revised version of the manuscript, two literature references related to electron poor C-4 unsubstituted 1,8-naphthalimides have been added as recommended Reviewer #3.

4) As Reviewer #3 recommended, Figure 3B is corrected in the revised version of the manuscript.

5) According to the recommendation of Reviewer #3, the Conclusions paragraph was rewritten giving exact findings from the results and discussion.

Round 2

Reviewer 1 Report

The manuscript has been revised and can be considered for publication.